# Nonalcoholic Fatty Liver Disease—A Novel Risk Factor for Recurrent *Clostridioides difficile* Infection

**DOI:** 10.3390/antibiotics10070780

**Published:** 2021-06-27

**Authors:** Lara Šamadan, Mia Jeličić, Adriana Vince, Neven Papić

**Affiliations:** 1School of Medicine, University of Zagreb, 10000 Zagreb, Croatia; lsamadan@mef.hr (L.Š.); avince@bfm.hr (A.V.); 2University Hospital for Infectious Diseases, 10000 Zagreb, Croatia; majdukovic@bfm.hr

**Keywords:** *Clostridioides difficile* associated disease, CDI, nonalcoholic fatty liver disease, NAFLD, NASH, recurrent disease

## Abstract

Recurrent *Clostridioides difficile* infections (rCDI) have a substantial impact on healthcare systems, with limited and often expensive therapeutic options. Nonalcoholic fatty liver disease (NAFLD) affects about 25% of the adult population and is associated with metabolic syndrome, changes in gut microbiome and bile acids biosynthesis, all possibly related with rCDI. The aim of this study was to determine whether NAFLD is a risk factor associated with rCDI. A retrospective cohort study included patients ≥ 60 years hospitalized with CDI. The cohort was divided into two groups: those who were and were not readmitted with CDI within 3 months of discharge. Of the 329 patients included, 107 patients (32.5%) experienced rCDI. Patients with rCDI were older, had higher Charlson Age–Comorbidity Index (CACI) and were more frequently hospitalized within 3 months. Except for chronic kidney disease and NAFLD, which were more frequent in the rCDI group, there were no differences in other comorbidities, antibiotic classes used and duration of antimicrobial therapy. Multivariable Cox regression analysis showed that age >75 years, NAFLD, CACI >6, chronic kidney disease, statins and immobility were associated with rCDI. In conclusion, our study identified NAFLD as a possible new host-related risk factor associated with rCDI.

## 1. Introduction

*Clostridioides difficile* infection (CDI), with increasing prevalence and mortality rates, is the leading cause of healthcare-associated diarrhea [1]. On hospital admission, 7% of patients are already colonized with *C. difficile* and another 21% become infected during hospitalization [2]. Furthermore, 20 to 30% of patients will develop symptomatic CDI recurrence within 2 weeks of completion of accurate therapy, and each episode increases the risk of future episodes by about 20% [3,4]. Therefore, recurrent *C. difficile* infections (rCDI) cause substantial impact on healthcare systems with limited and often expensive therapeutic options [5]. This requires identification of patients at high risk of rCDI. So far, major risk factors suggested to be associated with rCDI were age, use of antibiotics for non-*C. difficile* infection, gastric acid suppression and infection with a hypervirulent strain (NAP1/BI/027) [5]. Age is the most frequently reported risk factor for rCDI; the probability of rCDI is 27% in the age group between 18 and 64 and 58.4% in patients older than 65 years [4,6]. However, the literature data on other host risk factors are inconsistent and often contradictory. Severe underlying comorbidities, chronic renal disease, CDI severity, prolonged hospital stay, and nursing home residency were inconsistently reported as additional risk factors [5,6,7,8]. Recently, it was suggested that metabolic syndrome and its components, primarily obesity and diabetes mellitus (DM), might be associated with rCDI [8,9]. Metabolic syndrome is linked with changes in gut microbiome that might serve as protective intestinal flora for *C. difficile* germination and disease development [10].

Nonalcoholic fatty liver disease (NAFLD) is the most common chronic liver disease, affecting about 25% of the adult Western population [11,12]. NAFLD represents a spectrum of chronic liver pathologies, from simple steatosis to nonalcoholic steatohepatitis, cirrhosis, and hepatocellular carcinoma [11,12]. There is a close relationship between NAFLD and metabolic syndrome, clustering visceral overweight, dyslipidemia, insulin resistance and arterial hypertension [11,12,13]. Some consider NAFLD as the hepatic manifestation of metabolic syndrome. Importantly, it is well described that changes in the gut microbiome promote the development of NAFLD through inflammatory processes, insulin resistance and bile acids metabolism, all possibly related with increased susceptibility to CDI [14,15]. We have recently shown that NAFLD is an independent risk factor for in-hospital CDI in elderly patients treated with systemic antimicrobial therapy [16]. Similarly, a case–control study showed that hospitalized patients with CDI more frequently have NAFLD and metabolic syndrome, as compared to age- and gender-matched controls hospitalized for non-CDI diarrhea [17]. However, the question remains if NAFLD is a risk factor for CDI recurrence. Due to the high and increasing burden of NAFLD in the Western population, identification of NAFLD as a novel and common risk factor could improve current CDI management strategies.

Therefore, the aim of this retrospective cohort study was to determine whether NAFLD is an independent risk factor associated with CDI recurrence in elderly patients who are at the highest risk for increased mortality, hospitalization, complications and healthcare costs.

## 2. Results

A total of 329 patients were included in the study (196, 59.6% females with the median age of 77, interquartile range (IQR) 71–83 years). Among the entire cohort, 107 (32.5%) patients developed recurrent CDI within 90 days of index hospitalization discharge or by the end of index CDI therapy (total of 171 episodes). Of those, 67 patients developed rCDI by 28 days (a cumulative probability of recurrence of 20.7%, 95% CI = 13.5–28.9%) and 40 patients by 90 days (33.3%, 26.5–40.4%). Three, four and five or more documented episodes were registered in the studied period in 38, 11 and 5 patients, respectively.

Patients with rCDI were older (78 years, IQR 74–84 vs. 77 years, 71–81), had higher Charlson Age–Comorbidity Index (CACI) (6, IQR 5–7 vs. 5, IQR 4–6) and were more frequently hospitalized within 3 months (81.3% vs. 68.9%). Except for chronic kidney disease (CKD) (26.2% vs. 6.8%) and NAFLD (34.6% vs. 18.5%), which were more frequent in the rCDI group, there were no differences in other comorbidities, as presented in Table 1. Notably, the majority of patients with CKD in our cohort did not undergo chronic dialysis; two patients in CDI and four in rCDI group were in a dialysis program.

High proportions of patients in both groups were nursing home residents (37.4% vs. 33.3%). Regarding chronic medications, patients without recurrence were more frequently receiving statins (23.4% vs. 13.1%), while there were no differences in prescription of perioral anti-diabetic, metformin or insulin. The majority of patients in both groups were receiving histamine-2 receptor antagonist and/or proton pump inhibitors (47.7% vs. 43.7%).

Use of specific of antibiotic classes per patient and duration of therapy prior the first episode of CDI were similar; fluoroquinolones were most frequently prescribed (29.9% vs. 27.0%), followed by 3rd generation cephalosporins (20.5% vs. 13.5%) and amoxicillin/clavulanate (19.6% vs. 22.1%). The reasons for antibiotic prescription prior the first episode of CDI were similar in both groups—most commonly for urinary tract infections (28.9% vs. 30.2%), respiratory infections (26.2% vs. 23.4%) and skin/soft tissue infections (11.2% vs. 5.9%). However, there were 9 patients with rCDI (8.4%) and 42 (18.9%) with CDI, for whom the reason for antibiotic prescription was not clear.

On index hospitalization, the majority of patients presented with severe CDI, median ATLAS score of 5 (IQR 4–7) in both groups, and elevated inflammatory markers (C-reactive protein and white blood cells count), as presented in Table 2. There were no differences in APRI or FIB4 scores between groups; APRI score >1.5 had 4.0% of patients in CDI and 2.8% in rCDI group. FIB4 score >3.25 had 10.8% in CDI and 10.3% of patients in rCDI group.

There was no difference in the choice of CDI treatment between groups; 118 patients were treated with metronidazole, 182 with vancomycin and 29 received combined therapy. There were 39 patients considered metronidazole unresponsive who were switched to vancomycin. Importantly, the majority of our cohort received non-CDI antibiotics during hospitalization (62.6% in rCDI and 67.6% in CDI group). As shown in Table 2, the most commonly used antibiotics were piperacillin/tazobactam (23.4% vs. 18.5%), 3rd generation cephalosporins (23.4% vs. 21.6%) and carbapenems (10.3% vs. 14.9%). The reasons for non-CDI therapy were respiratory tract infection (15.5%), urinary tract infection (30.7%), sepsis (7.9%) and suspected bacteriaemia of gastrointestinal origin (6.1%). There were no differences between groups in the choice of antibiotic therapy, number of antibiotic classes prescribed or duration of antimicrobial therapy.

In order to identify potential risk factors for CDI recurrence, we performed a multivariable Cox regression analysis. After adjustment for potential cofounders, Charlson Age–Comorbidity Index >6 (Hazard ratio (HR) 1.97, 95% CI 1.32–2.92), age > 75 years (HR 1.88, 95% CI 1.20–2.97), NAFLD (HR 1.81, 95% CI 1.19–2.74, Figure 1), chronic kidney disease (HR 1.86, 95% CI 1.19–2.88) and immobility (HR 1.73, 95% CI 1.16–2.56) were associated with rCDI, as presented in Table 3. Chronic therapy with statins was associated with a decreased risk of rCDI (HR 0.24, 95% CI 0.11–0.52). Interestingly, some of the expected risk factors such as diabetes mellitus or obesity, nasogastric tube feeding and nursing home residency, as well as previous hospital admissions, choice of previous antibiotic therapy or concomitantly used antibiotics were not associated with rCDI in our model.

In addition, when NAFLD was combined with age > 75 years, chronic kidney disease and immobility, the risk of rCDI was even higher, as shown in Figure 2. Statin use was associated with lower rCDI in both patients with and without NAFLD (Figure 2, Panel d).

Finally, we report the demographic and clinical characteristics of patients with NAFLD. As previously shown, they had significantly higher incidence of rCDI (47.4% vs. 27.9%). They were younger than patients without NAFLD (median of 76, IQR 71–81 vs. 78, IQR 72–83 years), more commonly had BMI > 30 (38.5% vs. 9.6%), diabetes mellitus (41.0% vs. 22.3%) and hyperlipidemia (33.3% vs. 21.1%) (data not shown). Consequently, they were more frequently prescribed statins (26.9% vs. 17.9%), metformin (11.5% vs. 7.9%) and insulin (17.9% vs. 9.2%) (data not shown). There was no difference in other comorbidities, chronic medications or antibiotics used. The 1st episode disease severity was similar to patients without NAFLD (ATLAS score of 5, IQR 4–6 vs. 6, IQR 4–7, *p* = 0.103) (data not shown).

## 3. Discussion

Over the last decade, there has been a marked increase in the incidence and severity of CDI, with relapsing episodes now occurring at a higher frequency, especially in the elderly. Notably, recurrent CDI is associated with a significantly higher risk of death within six months after initial CDI treatment completion, compared with CDI patients who do not develop a recurrence [18,19].

In this retrospective cohort study, we found a significant association between recurrent CDI and NAFLD in elderly patients. Moreover, this appears to be independent of a number of potential confounders, specifically other components of metabolic syndrome.

While there are several well-established risk factors for primary infection with *C. difficile*, the studies examining risk factors for CDI recurrence had variable results depending on the population studied. To the best of our knowledge, none of them have analyzed the impact of NAFLD on CDI recurrence.

Surprisingly, the investigation of the role of NAFLD in bacterial infections has only recently been initiated. Although patients with NAFLD might have a higher risk for infections due to the concomitant presence of obesity or diabetes mellitus, few studies that included NAFLD in the analysis consistently showed its outcome impact independently of the metabolic syndrome components [20]. So far, this was suggested for community-acquired pneumonia, bacteriaemia of gastrointestinal origin, sepsis and urinary tract infections [21,22,23,24,25]. Recently, we have shown that NAFLD is a risk factor for in-hospital CDI development in elderly patients treated with systemic antimicrobial therapy [16].

The possible explanation of increased risk of CDI in NAFLD patients includes changes in intestinal microbiota linked to the development and progression of NAFLD [13]. While *Bacteroides* and *Bifidobacterium* play an important role in the mechanism, preventing colonization by *C. difficile*, patients with NAFLD were shown to have a relative decrease in the proportion of *Bacteroides* to *Firmicutes* [13,26,27]. Notably, NAFLD is now considered a multisystem disease due to the persistent low-level inflammation with impaired immune response that might predispose patients to a variety of infections [13,28].

In our cohort, DM and obesity were not associated with rCDI. The current medical literature is contradictory on the association between obesity and *C. difficile* infection. Two case–control studies that examined the association of CDI with BMI showed different results. While Bishara et al., based on data collected from 148 adult patients with CDI, showed an association of BMI with CDI (OR = 1.196 per 1-unit increase in BMI scale) [9], Punni et al. in their study on 189 patients did not [29]. There was no association of obesity with the risk of *C. difficile* infection among patients with ulcerative colitis and according to another study, obesity was even associated with decreased risk of CDI in hospitalized patients with pouchitis [30,31].

Meanwhile, several studies have shown that DM increases the risk of CDI recurrence, but none of them included NAFLD as a variable. A large Spanish cohort study showed a significantly higher incidence of CDI in DM patients, with an increasing trend between 2011 and 2015 [32]. In addition, patients with DM have significantly higher probability for hospital readmission due to CDI (adjusted OR of 3.79 to 5.46) and the development of severe CDI [8,33]. DM was also recognized as an independent risk factor in patients with toxigenic *C. difficile* colonization to develop *C. difficile*-associated diarrhea [7].

Interestingly, it was shown that metformin increases the *Bacteroidetes*/*Firmicutes* ratio; therefore, it may yield a protective effect against CDI in patients with DM [34]. A retrospective, case–control study compared CDI diabetic patients to diabetic patients without CDI and found metformin treatment to be associated with significant reduction in CDI (OR 0.58) [35]. This could be partially explained by the study that showed that metformin reduces vegetative cell growth of *C. difficile* in vitro, as well as ex vivo in human microbiome culture system [36]. Metformin modified human gut microbiome by decreasing *C. difficile* growth while increasing the growth of non-pathogenic Clostridium strains [36].

Other risk factors for rCDI in our study were age, significant comorbidities measured with Charlson Age–Comorbidity Index (CACI), chronic kidney disease and immobility.

Several studies examined the association of CKD with CDI. While the association of severe CKD requiring dialysis with CDI severity and mortality is clear, there are inconsistent data on if patients not undergoing dialysis are at higher risk for CDI [37,38,39,40]. We provide additional data that patients with CKD not requiring chronic dialysis are at increased risk for rCDI.

Meanwhile, antibiotics and proton pump inhibitors, which are well-known risk factors for initial CDI and rCDI, were not associated with rCDI in our study. This might be due to the characteristics of our cohort and widespread use of these medications, which might have predisposed the development of CDI in the first place.

Next, we found that statin use was associated with lower rCDI in patients both with and without NAFLD. Due to the global epidemic of cardiovascular diseases, obesity and metabolic syndrome, statins are considered as one of the most commonly used medications worldwide [41]. Other than their cholesterol-lowering effect, they also have an anti-inflammatory and immunomodulatory properties [42]. According to a recent meta-analysis of available data, the risk of developing CDI was approximately 25% lower in statin users compared with non-users [43]. However, this meta-analysis included eight observational studies with significant heterogeneity. There are no randomized control trials published so far. Although the exact mechanisms of risk decrease in CDI in statin users remains unknown, some studies demonstrated that statins have an influence on gut microbiota and could change its composition [44,45,46]. Vieira-Silva et al. recently identified statins as a key covariate of gut microbiome diversification [45]. In their study examining obesity-associated microbiota alterations, they showed that obesity is linked with intestinal microbiota configuration characterized by a high proportion of *Bacteroides*, a low proportion of *Faecalibacterium* and low microbial cell densities (the so called Bact2 enterotype) [45]. However, patients treated with statins had significantly lower Bact2 prevalence [45]. A systematic review of both human and animal data showed that statins modulate the gut microbiome, but the effect of change is unclear, probably due to the differences in populations studied [46]. It seems reasonable to speculate that modulation of the gut microbiome by statins might aid in the restoration of colonization resistance and lower recurrence of CDI. Alternatively, as statins have anti-inflammatory properties, their use may decrease the inflammatory response to *C. difficile* infection, which may lead to the decreased severity of CDI [47].

The major limitation of this study comes from its retrospective, monocentric design, and despite adjustment for a variety of demographics, comorbidities and medications, residual confounding may exist. The diagnosis of NAFLD was based on abdominal US, which is operator-dependent, and patients were not systematically screened for other causes of liver steatosis, but from data available in medical charts. Importantly, US has limited sensitivity and does not reliably detect steatosis when <20% or in individuals with high body mass index (BMI) (>40 kg/m^2^) and is inferior to MRI or CT scan for detection and grading of steatosis [48,49]. Next, while any potential effects of NAFLD should be interpreted in connection with metabolic syndrome, due to the retrospective design, we could not include waist circumference, type of dyslipidemia, levels of triglycerides or HDL cholesterol. However, we included diabetes mellitus and obesity (defined by BMI > 30 kg/m^2^), which might have the highest confounding impact, and have both been previously shown to be associated with rCDI. Since only a minority of patients had a significant risk of advanced fibrosis, as measured by APRI and FIB4 score, we were not able to determine the effect of advanced NAFLD on rCDI. Another limitation was the lack of data on *C. difficile* strain, which might be important since 027/BI/NAP1 strain has been associated with increased risk of CDI recurrence [50]. The study was designed to investigate the effect of NAFLD in elderly who are at the highest risk for rCDI and included only patients who were rehospitalized during a 3-month period. Therefore, patients diagnosed and treated for rCDI entirely as outpatients would not have been identified as having a recurrence. However, this would underestimate, rather than overestimate, the impact of NAFLD on patient outcomes.

Nevertheless, we report the first data examining the association of NAFLD with rCDI.

## 4. Materials and Methods

### 4.1. Study Design and Patients

This was a retrospective cohort study conducted at the University Hospital for Infectious Diseases Zagreb (UHID), Croatia, which is a national referral center for infectious diseases. We reviewed the hospital records of all adult patients hospitalized at UHID with a diagnosis of *Clostridium difficile* infection over a 5-year period (2016–2019). We included patients > 60 years diagnosed with the first episode of CDI. Patients who had a previous episode of CDI within three months before index hospitalization were excluded, as well as patients with known alcohol abuse and/or those diagnosed with chronic viral hepatitis or with a history of other known liver diseases. Next, only patients with performed ultrasonography examination to assess liver steatosis were included in study. During the period studied, a total of 841 patients were hospitalized with diagnosis of CDI. Of those, 730 were >60 years (total of 999 episodes of CDI). A total of 401 patients were excluded: 31 had CDI within 3 months, significant alcohol intake in 38, chronic viral hepatitis in 21, cirrhosis in 11 and hepatotoxic medications in 14 patients. There were 267 patients who did not have abdominal imaging, and 19 patients died during initial hospitalization. In the end, 329 patients were included in the study. The cohort was divided into two groups, those who were and were not readmitted with CDI within 3 months of discharge, as described in a flowchart (Figure 3).

### 4.2. Data Collection, Outcomes, and Definitions

We collected multiple variables that could be associated with patient outcomes. These variables included demographic data (age, sex and residence in nursing homes) and medical history (comorbidities (measured by Charlson Age–Comorbidity Index, CACI [51]), chronic medications, hospital admission within 3 months, administration of antibiotics within 90 days). CDI was defined as the presence of 3 or more unformed stools in 24 or fewer consecutive hours, confirmed with positive two-stage *C. difficile* stool tests (screening GDH test confirmed with toxin A/B PCR), according to current guidelines [52,53,54]. CDI severity was determined by ATLAS score calculation [55]. CDI treatment regimen, its duration and the use of other non-CDI antimicrobials were collected. Selected blood laboratory data at the admission were analyzed: C-reactive protein level, white blood cell count, platelet count, hemoglobin, blood urea nitrogen, serum creatinine, aspartate aminotransferase, alanine aminotransferase, gamma-glutamyl transferase, alkaline phosphatase, bilirubin and serum albumin concentration. In addition, as a surrogate marker of liver injury, APRI and FIB-4 score were calculated for all patients [56]. Using the CKD-EPI equation, we calculated estimated glomerular filtration rate (eGFR) [57]. The diagnosis of NAFLD was made based on the results of abdominal ultrasound and by the absence of a secondary cause of NAFLD, according to current guidelines [11,12]. The liver steatosis was assessed by ultrasound in all patients by an experienced radiologist, and defined as finding liver parenchyma with increased echogenicity and sound attenuation [49].

The primary study outcome was rCDI, defined as CDI occurring within 14–90 days of the initial CDI diagnosis date. Same as CDI, rCDI was defined based on compatible symptoms accompanied by a positive laboratory test.

### 4.3. Statistical Analysis

The clinical characteristics, laboratory and demographic data were evaluated and descriptively presented. We used Fisher’s exact test and the Mann–Whitney U test to compare the groups. All tests were two-tailed; a *p* value < 0.05 was considered statistically significant. Time to CDI recurrence was evaluated using the Kaplan–Meier method, and the comparison of CDI recurrence risk between patients with and without NAFLD was made using the log-rank test. Risk factors for the development of the rCDI were investigated using a univariate, and subsequently, a multivariable Cox regression model by estimating the hazard ratio (HR) and its 95% CI for the time from cure date of the primary CDI to the first episode of rCDI. Variables with *p* < 0.2 on univariate analysis or with clinical/biological plausibility were included in initial multivariable models. Multivariable Cox proportional hazards models were developed using backward elimination with *p* < 0.1 to retain variables in the model. Statistical analyses were performed using the GraphPad Prism Software version 9.1.1. (San Diego, CA, USA) and MedCalc version 20.008 (MedCalc Software, Ostend, Belgium).

## 5. Conclusions

In conclusion, we have shown that NAFLD is a novel host-related risk factor for recurrent CDI in elderly patients. This might be relevant for several reasons. Firstly, this highlights the need to include NAFLD as a variable in future studies, examining both CDI and rCDI. Secondly, patients with NAFLD might benefit from screening strategies, preemptive treatment or prophylactic measures, such as antibiotic prophylaxis that has recently been investigated [58,59]. The finding that statins reduce the risk of rCDI in both patients with and without NAFLD might be a novel treatment option that warrants further examination. Finally, there is growing evidence that immunological changes in patients with NAFLD might have a profound impact in the course of bacterial infections, the place where we have not been looking so far.

## Figures and Tables

**Figure 1 antibiotics-10-00780-f001:**
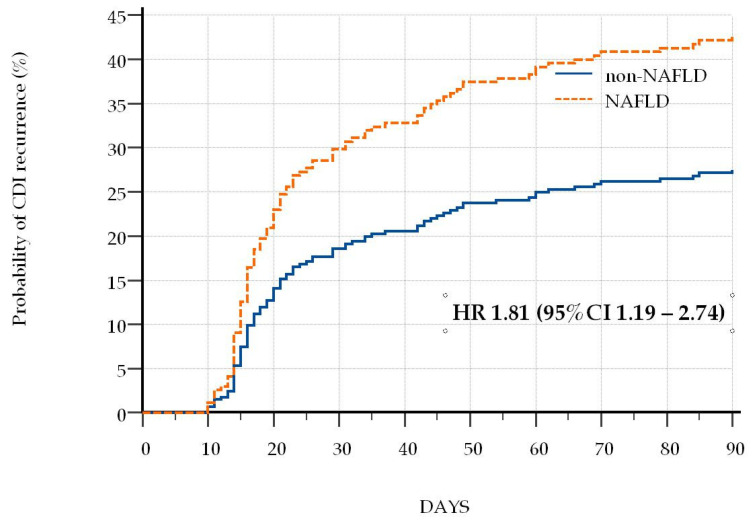
Kaplan–Meier curves and Cox proportional hazard ratios (HR) for recurrence of *Clostridioides difficile* infection in patients with and without nonalcoholic fatty liver disease.

**Figure 2 antibiotics-10-00780-f002:**
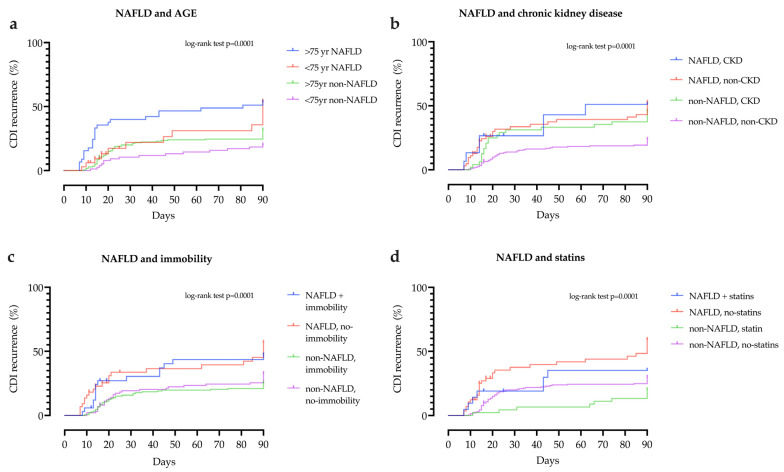
Kaplan–Meier curves for recurrence of *Clostridioides difficile* infection (CDI). Using the Kaplan–Meier method, a proportion of patients with CDI recurrence stratified by the concomitant presence of NAFLD and (**a**) age, (**b**) chronic kidney disease, (**c**) immobility and (**d**) statins during the follow-up period.

**Figure 3 antibiotics-10-00780-f003:**
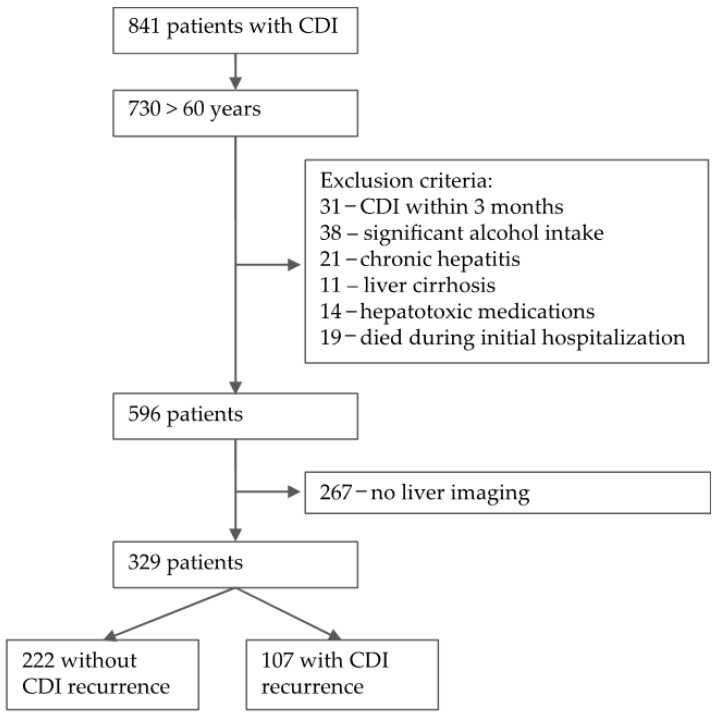
Study design flow chart.

**Table 1 antibiotics-10-00780-t001:** Baseline patients’ characteristics.

Characteristics	*Clostridioides difficile* Infection (*n* = 222)	Recurrent *Clostridioides difficile* Infection (*n* = 107)	*p*-Value ^a^
Age, median (IQR ^b^)	77 (71–81)	78 (74–84)	0.011
Female, No. (%)	135 (60.8%)	61 (57.0%)	0.549
Nursing Home Resident	74 (33.3%)	40 (37.4%)	0.536
Charlson Age–Comorbidity Index	5 (4–6)	6 (5–7)	<0.001
Hospital Admission within 3 months	153 (68.9%)	87 (81.3%)	0.018
**Comorbidities**			
Diabetes Mellitus	54 (24.3%)	34 (31.8%)	0.183
Arterial Hypertension	108 (48.6%)	59 (55.1%)	0.291
Cardiovascular Disease	57 (25.7%)	18 (16.8%)	0.092
Peripheral Vascular Disease	20 (9.0%)	9 (8.4%)	>0.999
Hyperlipidemia	54 (24.3%)	25 (23.4%)	0.891
Solid Tumor	22 (9.9%)	14 (13.1%)	0.451
Chronic Kidney Disease	15 (6.8%)	28 (26.2%)	<0.001
Chronic Obstructive Pulmonary Disease	51 (22.9%)	19 (17.8%)	0.316
Neurological Diseases	51 (22.9%)	27 (25.2%)	0.679
Obesity (BMI > 30 kg/m^2^)	23 (10.4%)	18 (16.8%)	0.109
Body Mass Index (BMI) ^c^	25 (23–28)	27 (24–30)	0.010
Nonalcoholic Fatty Liver Disease	41 (18.5%)	37 (34.6%)	0.002
**Use of Chronic Medications**			
Statins	52 (23.4%)	14 (13.1%)	0.028
Metformin	15 (6.8%)	13 (12.1%)	0.138
Other Perioral Anti-diabetic	13 (5.9%)	10 (9.3%)	0.255
Insulin	22 (9.9%)	15 (14.0%)	0.269
Histamine-2 Receptor Antagonist and/or Proton Pump Inhibitor	97 (43.7%)	51 (47.7%)	0.554
Immunosuppressive Agents	20 (9.0%)	7 (6.5%)	0.525
**Antibiotic Therapy before the 1st Episode of CDI**			
Fluoroquinolones	60 (27.0%)	32 (29.9%)	0.510
1st Generation Cephalosporins	1 (0.4%)	1 (0.9%)	0.545
2nd Generation Cephalosporins	18 (8.1%)	7 (6.5%)	0.824
3rd Generation Cephalosporins	30 (13.5%)	22 (20.5%)	0.108
4th Generation Cephalosporins	3 (1.3%)	1 (0.9%)	>0.999
Amoxicillin/Clavulanate	49 (22.1%)	21 (19.6%)	0.668
Piperacillin/Tazobactam	18 (8.1%)	9 (8.4%)	>0.999
Carbapenems	7 (3.1%)	5 (4.7%)	0.536
Macrolides	22 (9.9%)	8 (7.5%)	0.544
Clindamycin	12 (5.4%)	10 (9.3%)	0.237
Others	6 (2.7%)	6 (5.6%)	0.214

^a^ Fisher exact or Mann–Whitney U test, as appropriate; ^b^ IQR, interquartile range; ^c^ data available for 299 patients.

**Table 2 antibiotics-10-00780-t002:** Clinical, laboratory and treatment characteristics during the first episode of CDI.

Characteristics	*Clostridioides difficile* Infection (*n* = 222)	Recurrent *Clostridioides difficile* Infection (*n* = 107)	*p*-Value ^a^
**CDI Severity**			
Nonsevere	95 (42.8%)	45 (42.0%)	0.377
Severe	100 (45.0%)	54 (50.5%)	
Fulminant	27 (12.2%)	8 (7.5%)	
ATLAS Score	5 (4–7)	5 (4–7)	0.503
**Laboratory Findings on Admission**			
C-reactive Protein, mg/L	107.5 (54–172)	94.1 (54.6–147.0)	0.294
White Blood Cells Count, ×10^9^/L	14.5 (10.0–20.0)	13.2 (10.1–19.6)	0.487
Hemoglobin, g/L	120 (107–129)	120 (108–130)	0.851
Platelets, ×10^9^	251 (198–327)	256 (190–327)	0.859
Urea, mmol/L	8.1 (5.3–13.0)	7.1 (4.97–11.63)	0.256
Creatinine, μmol/L	103 (77.0–153.0)	109 (75–145)	0.890
Aspartate Aminotransferase, IU/L	19 (14–28)	19 (13–24)	0.146
Alanine Aminotransferase, IU/L	15 (10–23)	14 (10–22)	0.581
Serum Albumins, g/L	27.8 (23.9–32.4)	28.8 (24.8–33.9)	0.194
**CDI Treatment Regiment**			
Metronidazole	77 (34.7%)	41 (38.3%)	0.541
Vancomycin	122 (54.9%)	60 (56.1%)	0.906
Metronidazole + vancomycin	23 (10.4%)	6 (5.6%)	0.212
**Other Antimicrobials (not for *C.**difficile*)**			
Any Systemic Antibiotic	150 (67.6%)	67 (62.6%)	0.387
Fluoroquinolones	9 (4.0%)	8 (7.5%)	0.194
1st Generation Cephalosporins	2 (0.9%)	0	>0.999
2nd Generation Cephalosporins	2 (0.9%)	1 (0.9%)	>0.999
3rd Generation Cephalosporins	48 (21.6%)	25 (23.4%)	0.777
4th Generation Cephalosporins	2 (0.9%)	2 (1.9%)	0.598
Amoxicillin Clavulanate	18 (8.1%)	8 (7.5%)	>0.999
Piperacillin/Tazobactam	41 (18.5%)	25 (23.4%)	0.307
Carbapenems	33 (14.9%)	11 (10.3%)	0.301
Macrolides	5 (2.2%)	5 (4.7%)	0.304
Clindamycin	1 (0.4%)	1 (0.9%)	0.545
Others	8 (3.6%)	5 (4.7%)	0.763
No. of Antibiotic Classes Used per Patient	1 (0–3)	1 (0–3)	0.889

^a^ Fisher exact or Mann–Whitney U test, as appropriate.

**Table 3 antibiotics-10-00780-t003:** Multivariable Cox regression analysis of risk factors for the development recurrent *Clostridioides difficile* infection.

Variable	Hazard Ratio	95% CI	*p*-Value
Age > 75 Years	1.88	1.20 to 2.97	0.006
Charlson Age–Comorbidity Index (CACI) > 6	1.97	1.32 to 2.92	<0.001
Immobility	1.73	1.16 to 2.56	0.006
Nonalcoholic Fatty Liver Disease	1.81	1.19 to 2.74	0.005
Chronic Kidney Disease	1.86	1.19 to 2.88	0.006
Statins	0.24	0.11 to 0.52	<0.001

The strength of association was expressed as hazard ratio (HR) and its corresponding 95% confidence interval (CI). The area under the ROC curve in the fully adjusted model was AUC 0.72 (95% CI 0.66 to 0.77).

## Data Availability

The datasets generated and/or analyzed during the current study are available from the corresponding author on reasonable request.

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
