# Peer review of "Nonalcoholic Fatty Liver Disease—A Novel Risk Factor for Recurrent Clostridioides difficile Infection"

_antibiotics, 2021, doi:10.3390/antibiotics10070780_

Round 1

Reviewer 1 Report

The paper authored by Šamadan and colleagues summarizes the vital and interesting topic of factors associated with a higher risk of recurrent C. difficile infection. However, there are some major issues that need to be resolved before further consideration.   Major issues: 1. Authors focused their paper on the presence of NAFLD as a key risk factor for recurrent C. difficile infections. However, the assumed association has several theoretical and methodological weaknesses. First of all, this regards the pathophysiological nature of NAFLD. A long ongoing debate is on whether NAFLD should be regarded as a reason or consequence of cardiometabolic disorders. Therefore, never any potential “effects” of NAFLD should be interpreted in disconnection with metabolic syndrome (especially in a cross-sectional setting). The authors made a firm conclusion (Line 166) that identified NAFLD association was independent of co-presence of metabolic syndrome.    a. My concern is that metabolic syndrome and all of its components were not studied in this investigation.  Authors extracted and reported central obesity and ambiguously defined hyperlipidemia and diabetes mellitus. How about other components such as raised systolic and diastolic blood pressure (or hypertension treatment), waist circumference (abdominal obesity), specification of dyslipidemia (i.e. high TAG and low HDL)? b. Second concern is regarding the stepwise variable selection into the model. This is, of course, one of the frequently used approaches to select the best fitting model. However, stepwise variable selection requires an extremely careful understanding of the nature of variables entered into the model. Here we expect a high correlation (or co-occurrence to be precise) between NAFLD and metabolic syndrome components (authors somehow reference this in lines 139 and 146). In a situation of high multicollinearity between variables, such features like any metabolic syndrome component can be regarded as redundant and omitted from the model. In such a situation, variables with greater effect size, such as NAFLD (as it is a proxy for clustered metabolic disorders) may remain in the model. But such a situation does not provide high certainty for the independence of association.   c. A solution for the present situation would be to consider the formal definition of metabolic syndrome (e.g., IDF one) and its components (either as biochemical markers and treatment). Further, they should carefully assess the interrelation between NAFLD and metabolic syndrome and later construct model(s), which could support their conclusions on NAFLD. I cannot further refer to these results nor comment on discussion and conclusions in the current situation.   2. Authors state that a “relatively large” study sample is a strength. However, there is no power calculation to support this. Consequently, power calculation should be presented to give a rationale for the studied sample – Maybe the inability to detect differences in obesity or DM prevalence between CDI and rCDI was due to insufficient statistical power?   3. Line 284-295 – The study sample creation is not clear. 730 subjects were aged>60 years. 401 subjects were excluded due to other criteria – 730-401 = 329 patients – which is named as the final study sample. But 267 patients had no ultrasound imaging. Before authors wrote, “Next, only patients 249 with preformed ultrasonography examination to assess liver steatosis were included in 250 study”. Does it mean that final sample size should be 62 patients?? Please clarify and make the description more readable (considering adding a flowchart).   4. NAFLD evaluation using ultrasound is inferior to other methods such as CT or MRI. Therefore, considering that the authors selected NAFLD/liver fat as a primary focus of the study, methods for evaluation should be described in detail (e.g., cut-off point).   5. I would encourage authors to use Cox regression models to assess the time to recurrence of CDI. Kaplan-Meier curves can be shown as a graphical presentation of results. However, Cox models would allow for simultaneous adjustment for other potential confounders.    Minor issues: 1. Authors inconsistently use both British (i.e., Line 50 – dislipidaemia) and American spelling (i.e., Line 243 – center). Please proofread the manuscript carefully before submission of a revised version. 2. Line 28 – typo in the word “admission”. 3. Line 28 – inconsistent form with further text – “health-care” and “healthcare” 4. Line 31,33 – rCDI abbreviation in introduced twice. Please revise. 5. Line 36 – “for non-C.difficile” is the word infection missing? 6. Line 38 – missing “the” before “age group” 7. Line 41-42 “were inconsistently reported 41 as additional risk factors” – there are missing references here. 8. Line 43 – “primary” – adverb “primarily” seems to be fit better to the context. 9. Line 48 – the abbreviation “NASH” is introduced but never later used in the manuscript. Moreover, it is one of the study keywords, but it is not the focus of the study. Please revise. 10. Line 52 – typo in the word “Importantly” 11. Line 53 – should be “promote the development of NAFLD” 12. Line 55 – should be “have shown” not “have showed” 13. Line 60 – should be “in the western population” 14. Line 63 – study - please be consistent with name of study design you use. A retrospective study can be even a case-control study. Please write consistently that it was a retrospective cohort study. 15. Line 69 and later throughout the manuscript – please be consistent with the precision of rounding to decimal places. There is no goal of presenting a second decimal place for percentage in case of a sample size of ~300. Rounding to 0 or 1 decimal place is sufficient. 16. Line 76 – IQR abbreviations should be introduced in the first place it is used. 17. Line 77 - Charlson Age-Comorbidity Index (CACI) is consistently further called as age adjusted Charlson Comorbidity Index (ACCI). 18. Table 1 – Consider “Use of chronic medications” . Avoid use of “diabetic” (also in line 89) as a noun. Here it should be “anti-diabetic”. 19. Line 90-91 again abbreviations introduced are not used anywhere else (H2RA, PPI). 20. Line 92 – not number of classes but rather use of specific classes was analysed, correct? 21. Line 97 and later- authors inconsistently report percentage values in one place as (%1 , %)2 and in another as (number1, %1, number2 %2). Please be consistent.  22. Line 103 – please be consistent not leucocytosis (as a categorical state) but continuous WBC was reported. 23. Data for results described in lines 95-99 is not shown. Please note that as data not shown. 24. Line 161 – should be “with a significantly higher”. 25. Line164 – missing comma after “retrospective study”. 26. Line 187 – should be “the association”. 27. Line 189 and later – when referencing first authors (et al.) initials should not be reported.  28. Line 198 – should be “a significantly higher”. 29. Line 213 – should be “the risk” 30. Line 243 – should be “a national referral center”. 31. Line 245 – There are many abbreviations introduced in the Methods section but not used at all in the manuscript (e.g., biochemical markers which were before in tables called by full name). Some of the abbreviations are re-introduced (i.e., CDI), and some are changed (i.e., CACI and ACCI). I understand that the methods section placed at the end of the manuscript is tricky, but please be consistent with the style. 32. Line 263, 274 – Use of Charlson Comorbidity Index as well as FIB-4 and APRI score should be referenced such as ATLAS score.

Reviewer 2 Report

This article is an important paper that points out NAFLD as a risk for recurrent CD. The paper's content is well-organized and generally properly examined, although the statement for selecting patients and the statistical methods could be improved.

Major,

  1. It is stated that the patient was diagnosed with NAFLD by the echo. Is it correct my understanding that all patients admitted with CD underwent echo, or were the cases listed in the medical record as having comorbid NAFLD at the time of admission? Please clarify as this is an important point.
  2. You stated that APRI and FIB4 were calculated, but no data is presented. Please consider and show whether there is a relationship between the severity of NAFLD and CD recurrence.
  3. In Table1. 2, BMI must be presented.
  4. Although it is stated that the potential cofounders were adjusted in the multivariate analysis, it is essential to specify the variables used in the multivariate analysis.
  5. Were all patients hospitalized at the time of CD recurrence? Did any of them receive outpatient treatment? What about the possibility that some of the patients have been hospitalized and treated in other hospitals?
  6. Interestingly, you describe the association between statins and the development of CD. You discuss the mechanism simply as “an anti-inflammatory effect,” but I would like to know a more detailed molecular biology mechanism, citing previous studies in a discussion.

Minor,

Please improve the wording of the grouping in Figure 1b, as it is difficult to understand.

Reviewer 3 Report

This manuscript is interesting in terms of the relation between metabolic diseases and infectious diseases because both of them is associated with gut microbiota. However, I have some questions about your study. 

  1. Therapeutic agents such as statin and metformin which used to treat metabolic diseases may have impact on the gut microbiome status of (especially) CDI patients. Are there metagenome data in CDI patients with NAFLD treated or not treated with statin?
  2. It seems necessary to describe the reasons that statin lowered CDI recurrence. 
  3. As well known, NAFLD is induced from obesity. So Most NAFLD patients have more than one metabolic disease such as obesity or diabetes together. Can it be seen as a result of only NAFLD, which excludes the obesity? To verify the effect of NAFLD on the rCDI, the study using non-obese NAFLD patients seems appropriate.   
  4. Although there is significance between CKD and rCDI, why did you not elaborate on CKD as a risk factor? 

Round 2

Reviewer 1 Report

I want to thank authors for efforts made to revise the manuscript. There are still a few issues to address:

  1. Line 128-136 - please report estimates (HRs) with 95% CIs for all identified factors.
  2. Line 132 - HRs not ORs are reported! Please be consistent and revise.
  3. Line 150-155 - Such a description of Kaplan-Meier curves is not needed anymore as you report Cox models. Please refer to them in an earlier paragraph on Cox regression results - just in one or two sentences.
  4. Line 156-165 - An information (data not shown) is still missing! Please revise that.
  5. Figure 1 and following + Tables - Please make sure that all titles are self-descriptive (avoid abbreviations). 
  6. Table 1 and following - Please be consistent and make sure that all abbreviations in tables are explained in footnote - e.g., CDI and rCDI in Table 1 are not.
  7. Table 1 and everywhere else - Precision of 3 digital places will be enough for P-values (instead of 4 digits as it is). Please revise those.
  8. Regarding "white blood cells" - Please make sure that whenever applicable you follow this term with the word "count".

Author Response

We would like to thank the reviewer for the careful consideration and all the suggestions that helped to improve our manuscript. We have addressed all the issues raised by the reviewers in the revision.  

1. Line 128-136 - please report estimates (HRs) with 95% CIs for all identified factors.

Answer: HR and 95% CI are now added in the text.  

2. Line 132 - HRs not ORs are reported! Please be consistent and revise.

Answer: Corrected.

3. Line 150-155 - Such a description of Kaplan-Meier curves is not needed anymore as you report Cox models. Please refer to them in an earlier paragraph on Cox regression results - just in one or two sentences.

Answer: A following sentence was added in earlier paragraph: “In addition, when NAFLD was combined with age > 75 years, chronic kidney disease and immobility , the risk of rCDI was even higher, as shown in Figure 2. Statin use was associated with lower rCDI in both patients with and without NAFLD (Figure 2, Panel d).”  

4. Line 156-165 - An information (data not shown) is still missing! Please revise that.

Answer: “Data not shown” is added.

5. Figure 1 and following + Tables - Please make sure that all titles are self-descriptive (avoid abbreviations). 

Answer: corrected.  

6. Table 1 and following - Please be consistent and make sure that all abbreviations in tables are explained in footnote - e.g., CDI and rCDI in Table 1 are not.

Answer: corrected.  

7. Table 1 and everywhere else - Precision of 3 digital places will be enough for P-values (instead of 4 digits as it is). Please revise those.

Answer: corrected.  

8. Regarding "white blood cells" - Please make sure that whenever applicable you follow this term with the word "count".

Answer: corrected.  

Reviewer 2 Report

Appropriate corrections have been made. I am honored to have the opportunity to review this paper. I sincerely wish you success.

Author Response

We would like to thank the reviewer for the careful consideration and all the suggestions that helped to improve our manuscript. 

Reviewer 3 Report

In the last review, I raised some questions, and the authors responded to them one by one. I have no further suggestion. Sincerely, I look forward to your next research.

Author Response

(The authors gave the same response as above.)
